# Higher predicted type 2 diabetes risk is associated with worse mental health and self-rated general health among adults without known diabetes in Germany – Results of the nationwide population-based study GEDA 2022

**Laura Neuperdt**\*, **Yong Du, Julia Nübel**, **Ulfert Hapke, Lena Walther,**
**Gert B. M. Mensink**, **Almut Richter**, **Christa Scheidt-Nave, Jens Baumert,**
**Christin Heidemann**

Robert Koch Institute, Department of Epidemiology and Health Monitoring, Berlin, Germany

\* NeuperdtL@rki.de

## Abstract

### Objective

There is increasing evidence that the link between type 2 diabetes (T2D) and mental health may be partly due to shared modifiable risk factors. The present study examined whether a higher predicted T2D risk is associated with poorer self-rated general health (SRH) and mental health (SRMH) as well as depressive and anxiety symptoms among adults without diagnosed diabetes.

### Methods

Analyses are based on cross-sectional data from 4,909 adults (18 + years) without known diabetes who participated in the nationwide telephone interview survey German Health Update (GEDA) 2022. The predicted 5-year T2D risk (in %) was assessed with the German Diabetes Risk Score (GDRS) and categorized into very low (<2%), low (2- < 5%), elevated (5- < 10%) and high (≥10%) risk groups. Poisson regression was used to calculate prevalence ratios (PRs) of SRH, SRMH, depressive and anxiety symptoms according to T2D risk categories and models were adjusted for sex, age, education, region, living alone, and social support.

### Results

Predicted T2D risk (95% confidence interval (CI)) was very low for 60.8% (58.7–63.0%), low for 15.7% (14.2–17.2%), elevated for 10.7% (9.5–12.1%), and high for 12.8% (11.5–14.2%) of adults. Compared to those with very low T2D risk, participants at high T2D risk were less likely to report very good/good SRH (PR; 95% CI: 0.65;

**Data availability statement:** The data for this study cannot be made publicly available because informed consent from study participants did not cover public deposition of data. However, the data underlying the findings is archived in the Research Data Centre at the Robert Koch Institute (RKI) and can be accessed by researchers on reasonable request. On-site access to the data set is possible at the Secure Data Centere of the RKI's Research Data Centre. Requests should be submitted to the Research Data Centre (FDZ), Robert Koch Institute, Berlin, Germany via email (fdz@rki.de). Further information about access and request procedures can be found on the FDZ website (https://www.rki.de/EN/News/Publications/Research-data/Scientific-Use-Files/scientific-use-files-node.html).

**Funding:** This work was supported by a grant from the German Center for Diabetes Research (DZD) funded by the German Federal Ministry of Education and Research (grant number: HMGU2022Z5) and grants from the German Federal Ministry of Health within the framework of the National Diabetes Surveillance project at the Robert Koch Institute (grant numbers: 2522DIA700 and 2523DIA002). The survey GEDA 2022 was funded by the Robert Koch Institute and the Federal Ministry of Health in Germany. The funders had no role in study design, data collection and analysis, decision to publish, or preparation of the manuscript.

**Competing interests:** The authors have declared that no competing interests exist.

0.56–0.75) or excellent/very good SRMH (0.65; 0.51–0.81) and more likely to have depressive (2.48; 1.70–3.63) or anxiety symptoms (2.50; 1.54–4.05).

## Conclusion

Findings underline that physical and mental health should be considered together in the context of prevention and health promotion strategies.

---

## Introduction

Diabetes mellitus is an etiologically heterogeneous chronic disorder of glucose metabolism characterized by elevated plasma glucose levels due to impaired insulin secretion, insulin resistance, or a combination of both [1]. Based on data provided by the NCD Risk Factor Collaboration, 828 million people aged 18 years and older had diabetes worldwide in 2022 [2]. In Germany, the prevalence of self-reported diagnosed diabetes among adults was 10.3% (women: 9.0%, men: 11.6%) according to data collected in the nationwide population-based study Health in Germany 2024 [3]. Diabetes, if not treated properly, can lead to a range of diabetes-specific complications and comorbidities affecting multiple organs [4]. Beyond severe physical comorbidities, diabetes is associated with poorer self-rated health (SRH) [5] as well as common mental health disorders, such as depression [6–9] and anxiety [10–12]. The most common type of diabetes is type 2 diabetes (T2D) [13]. The pathogenesis of T2D involves a complex interplay of genetic predisposition, lifestyle, and environmental factors [14,15]. Identifying and reducing modifiable risk factors for T2D at an early stage of metabolic disturbances has the potential to prevent or delay the development of T2D.

There is increasing evidence that several major modifiable risk factors of T2D are also risk factors of poor SRH and mental health. Addressing these factors in health promotion and primary prevention could therefore have benefits far beyond T2D prevention. For example, physical activity was found to be associated with better self-rated mental health (SRMH) [16] and lower risk of depression [17,18]. A healthier dietary pattern seems to be associated with a reduced likelihood of depressive symptoms [19,20]; and high ultra-processed food consumption was shown to be related with a higher likelihood of depressive symptoms [21–23]. Importantly, different lifestyle-related risk factors can interact with each other [24] and can cause synergistic effects [25]. So far, only a few studies have analysed the association of multiple lifestyle-related factors in combination with either SRH [26–28] or specific mental health outcomes, e.g. with perceived chronic stress [29]. Results from these studies suggest that an overall healthier lifestyle characterized by multiple factors is associated with better health outcomes. However, an integrative approach that systematically examines the association between an overall risk profile of multiple health-related factors combined and self-rated physical and mental health as well as depressive and anxiety symptoms within the German population has been lacking.

In this study, we address this research gap by investigating associations between T2D risk, measured by the German Diabetes Risk Score (GDRS), and general and

mental health in a recent nationwide population-based sample of adults in Germany without a history of diabetes. The GDRS, which is a validated non-invasive instrument suitable for estimating the 5-year probability of developing T2D in non-clinical settings [30–32], integrates information on major behaviour-related health determinants (physical activity, dietary habits, smoking, waist circumference) as well as medical history of hypertension and family history of diabetes. It is also predictive of non-fatal and fatal cardiovascular disease [33] and can hence be regarded as a broader marker of cardiometabolic risk. The present study therefore offers a novel and comprehensive perspective on the association between T2D risk and mental health by integrating multiple health dimensions. We hypothesized that individuals with a higher predicted 5-year T2D risk are less likely to have very good/good SRH and excellent/very good SRMH and are more likely to have depressive and anxiety symptoms.

## Methods

### Study design and study population

The present study uses data from the 2022 wave of GEDA ('German Health Update'), which is a series of nationwide cross-sectional population-based health interview surveys among the German-speaking resident adult population. GEDA 2022 was conducted using a fully structured, computer-assisted telephone interview (CATI). Participants were randomly sampled using the same dual-frame landline and mobile sampling procedure previously employed (methodological details described elsewhere) [34]. GEDA 2022 comprised monthly samples and included a core health questionnaire as well as four health modules, each directed at a subsample. Overall, 33,149 adults 18 years of age and older participated in GEDA 2022 in ten time periods from 09/02/2022 to 14/01/2023. The present analyses were restricted to data from June 2022 to January 2023 (n = 5,796), as the relevant questions for assessing the GDRS were only included in this period in one of the health modules. Adults with a self-reported diabetes diagnosis (n = 597), with missing information on diabetes history (n = 7) or missing information on one of the GDRS variables (n = 283) were excluded, resulting in a sample of 4,909 adults for the present analyses. The study was conducted in adherence to the principles of the Helsinki Declaration [35] and the guidelines provided by the Federal Commissioner for Data Protection and Freedom of Information in Germany. It was approved by the Charité, Universitätsmedizin Berlin local ethics committee (No. EA2/201/21). All participants in the study gave their verbal consent to the storage of their data and its anonymised transfer to the Robert Koch Institute (RKI) ('informed consent'). Consent was based on the free decision of the person called, who was informed of the voluntary nature of participation and the anonymised processing of the data. Verbal consent was obtained after the interviewer had read aloud the data protection consent form; the respondent's agreement was documented electronically by recording the value "1" in the survey database, and the interviewer acted as the witness of the consent procedure. This procedure, including its documentation, was approved by the Charité ethics committee (No. EA2/201/21), which served as the competent Institutional Review Board (IRB). This complies with Article 7(1) General Data Protection Regulation ("Datenschutz-Grundverordnung, DSGVO") regarding the verifiability of consent by the responsible site. If the respondent did not consent, the interview was not conducted. The use of verbal consent in accordance with the procedure described is an established standard practice in nationwide telephone health surveys in Germany.

### T2D risk

The predicted 5-year risk of developing T2D was estimated using the German Diabetes Risk Score (GDRS) [36]. The assessment tool was developed by the German Institute of Human Nutrition Potsdam-Rehbruecke (DIfE) [37] and has been validated in population-based cohort studies of adults in Germany [31,38]. For the present analysis, the updated and validated simplified GDRS version was applied [31,36]. This version considers the following lifestyle-related and biological risk factors: chronological age, body height, waist circumference, physical activity, former and current smoking and average number of cigarettes, cigarillos, cigars, or pipes smoked per day, medical history of hypertension, family history

of diabetes (diabetes diagnosis in one or both biological parents or in at least one sibling), intake of red meat (beef, pork or lamb) and wholegrain products (wholegrain bread, rolls or muesli), and coffee consumption [39]. Waist circumference could not be assessed in this survey and therefore was estimated based on self-reported body weight (in kg), body height (in cm), age, and sex using the equation derived from linear regression models in a previous analysis (men: $R^2 = 0.86$; women: $R^2 = 0.82$) [37]. In this sample, the estimated sex-specific mean waist circumference was 95.9 cm in men (95% CI 95.0–96.8; margin of error ±0.90 cm) and 84.7 cm in women (95% CI 83.9–85.5; margin of error ±0.81 cm). The calculated T2D risk was categorized into "low risk (<2%)", "still low risk (2% to <5%)", "elevated risk (5% to <10%)", and "high risk (≥10%)" as previously applied [39]. For a better understanding, the category "low risk" was labelled as "very low risk" and "still low risk" was labelled as "low risk".

### Health outcomes

The internationally established indicator SRH is part of the Minimum European Health Module (MEHM) [40] and was assessed and operationalised in line with the recommendation of the World Health Organisation (WHO) [41]. Participants were asked: "How is your health in general?". The five response options ("very good", "good", "fair", "bad","very bad") were dichotomised into the established categories "very good/good" and "fair/bad/very bad".

SRMH was measured using an established single item [42]. Participants were asked: "How would you describe your overall mental health?". The five response options ("excellent", "very good", "good", "fair", "bad") were combined into "excellent/very good" and "good/fair/bad" in accordance with its application as an indicator of positive mental health in mental health surveillance systems of different countries [43–45].

Depressive symptoms were assessed using the Patient Health Questionnaire-2 (PHQ-2), which is a validated two-item screener widely used in population-based studies [46]. The PHQ-2 measures the frequency of depressed mood and anhedonia over the past two weeks. For each item, the response options were "0=not at all", "1=several days", "2=more than half the days", and "3=nearly every day" (scoring range: 0–6). The established cut-off indicating potentially clinically relevant depressive symptoms is ≥ 3 [47].

Anxiety symptoms were assessed using the validated Generalised Anxiety Disorder-2 (GAD-2) [46], which addresses two symptoms of generalised anxiety. The response categories and scoring are identical to the PHQ-2. The established cut-off indicating potentially clinically relevant anxiety symptoms is ≥ 3 [48].

### Covariates

The following covariates were considered based on observed associations with T2D risk [39] as well as with SRH [49,50] and mental health [42,51-53] and the availability of information in the study GEDA 2022. Age, sex and educational level defined by the Comparative Analysis of Social Mobility in Industrial Nations (CASMIN) classification system (categorized into low, middle and high education [54]) were included as sociodemographic covariates. Living alone (vs. not living alone) and social support measured with the Oslo-3 Social Support Scale (OSS-3, range 3–14, categorized into 3–8: poor social support, 9–11: moderate social support, and 12–14: strong social support) [55,56] were considered as potential social determinants. Residential region comprised North-East, Central-East, North-West, Central-West and South of Germany [57].

### Statistical analysis

The complex survey structure of the GEDA data was accounted for by using the survey procedure for all analyses within the statistical software Stata (version 17). A weighting factor correcting for deviations from the population structure with respect to the distribution of age, sex, federal state and district type as of 31 December 2020 as well as the distribution of education according to the International Standard Classification of Education (ISCED classification) in the 2018 Micro-census was used across analyses. The survey design was specified with the Stata option "svyset [pweight=wQS_tmod]";

primary sampling units (PSUs) or strata were not used within the telephone survey design, thus clustering and stratification did not have to be accounted for. Variances were estimated using Taylor linearization, which is the default in Stata's survey procedures.

Bivariate associations between T2D risk (categorical independent variable) and the health outcomes SRH, SRMH, depressive and anxiety symptoms were examined using the Rao-Scott chi-square test. Likewise, the relationship between age group and health outcomes was examined in an additional analysis as it is known that older people are at higher risk of T2D [39,58]. Percentages with 95% confidence intervals (CI) and p-values based on two-sided tests are reported.

Poisson regression [59] with a robust (sandwich) variance accounting for the survey design applying the Poisson command was used to examine the association between T2D risk categories and general and mental health adjusted for sociodemographic and social covariates. The decision to use Poisson regression was based on the relatively high prevalences that were observed in health outcomes. Prevalence ratios (PRs) with 95% CI were estimated using modified Poisson regression models with a log link, implemented with the "svy:" prefix to account for the complex survey design [60]. Separate models were fitted including T2D risk categories as the independent variable (reference "very low risk": <2%") and the individual health outcomes as dependent variables. For each outcome variable, we fitted four models, starting with T2D risk only (model 1), then stepwise adding sex (model 2), continuous age (model 3), and educational level, living alone, social support, and region (model 4). To avoid over adjustment, we did not include individual GDRS components as separate covariates. Because age is part of the GDRS, but is associated with depressive symptoms and anxiety in an opposite direction compared to the GDRS, we implemented the stepwise model sequence as described above with adding age not before model 3 to specifically assess how age adjustment changes the T2D risk-outcome association.

Missing values for educational level (n = 12), living alone (n = 8) and social support (n = 130) were excluded from model 4. Considering the overall high proportion of missing values in model 4, we performed a sensitivity analysis using multiple imputation for missing covariate data (educational level, living alone and social support) as well as for outcome variables. Another sensitivity analysis was performed modelling the natural log-transformed T2D risk as a continuous independent variable.

Further, we tested whether the effects of T2D risk on general and mental health outcomes were modified by age or sex by adding the respective interaction term (product term) of T2D risk (natural log-transformed) with age (continuous) or sex into the models. Finally, we calculated pairwise Pearson correlations between T2D risk, age, and all outcomes accounting for the survey design with Stata command "pwcorr" to test for the possibility of a suppression effect of age. A suppression pattern arises when the exposure and a covariate are correlated but relate to the outcome in opposite directions, so that adjusting for the covariate increases the absolute exposure-outcome association [61].

## Results

### Characteristics of the study population and T2D risk

The mean age of the 4,909 participants was 50.2 years (95% CI 49.4–51.1). The geometric mean of T2D risk was 1.15% ((95% CI 1.06–1.24), median = 1.21% (interquartile range, IQR: 0.27, 4.81)).

Table 1 summarizes the main characteristics of the study population across categories of T2D risk. Overall, 60.8% of adults were at very low risk of T2D and 15.7% were at low risk, while 10.7% had an elevated risk and 12.8% had a high T2D risk. The proportion of very low T2D risk was higher among women than men, among the younger than the older age groups, and among the high and middle educational groups than the low educational group, whereas the proportions with high risk were lower among the respective groups. Further, the proportion of very low T2D risk was lower among participants living alone than those living with others and among participants with poor social support than those with strong support, whereas the proportions with high risk were higher in the respective groups. The proportions within the risk categories did not differ significantly between the proportions of the regions .

**Table 1. Characteristics of the study population of adults without diabetes by categories of T2D risk (n = 4,909).**

| | | n | Categories of T2D risk | | | |
|---|---|---|---|---|---|---|
| | | | Very low risk (<2%) | Low risk (2% to <5%) | Elevated risk (5% to <10%) | High risk (≥10%) |
| | | | % (95% CI)[a] | % (95% CI)[a] | % (95% CI)[a] | % (95% CI)[a] |
| Overall | | 4,909 | 60.8 (58.7-63.0) | 15.7 (14.2-17.2) | 10.7 (9.5-12.1) | 12.8 (11.5-14.2) |
| Sex | Female | 2,640 | 66.1 (63.4-68.8) | 14.5 (12.9-16.3) | 10.0 (8.3-11.9) | 9.4 (7.9-11.1) |
| | Male | 2,269 | 55.3 (52.0-58.6) | 16.9 (14.6-19.5) | 11.4 (9.7-13.5) | 16.3 (14.2-18.7) |
| Age | 18-44 years | 1,174 | 93.1 (90.4-95.0) | 3.9 (2.5-5.9) | 2.5 (1.4-4.4) | 0.6 (0.1-2.3) |
| | 45-64 years | 1,922 | 55.8 (52.2-59.3) | 21.4 (18.7-24.5) | 10.6 (8.4-13.3) | 12.2 (9.9-14.8) |
| | ≥65 years | 1,813 | 11.3 (9.6-13.2) | 27.9 (24.9-31.2) | 25.4 (22.5-28.5) | 35.4 (32.0-39.0) |
| Educational level | Low | 754 | 37.3 (32.0-42.8) | 21.3 (17.4-25.7) | 17.0 (13.8-20.8) | 24.5 (20.6-28.9) |
| | Middle | 2,161 | 67.7 (64.9-70.4) | 14.4 (12.6-16.4) | 8.9 (7.4-10.8) | 9.0 (7.5-10.6) |
| | High | 1,982 | 71.1 (68.5-73.6) | 12.0 (10.5-13.8) | 8.0 (6.7-9.5) | 8.9 (7.6-10.3) |
| Region in Germany | North-East | 519 | 55.7 (48.5-62.6) | 17.8 (13.4-23.2) | 12.2 (8.4-17.4) | 14.4 (10.4-19.6) |
| | Central-East | 495 | 56.7 (50.2-63.1) | 17.1 (13.0-22.0) | 10.2 (6.9-14.8) | 16.0 (11.7-21.4) |
| | North-West | 809 | 60.2 (54.8-65.4) | 17.9 (14.1-22.5) | 11.4 (8.6-14.9) | 10.5 (8.0-13.7) |
| | Central-West | 1,661 | 59.5 (55.7-63.2) | 16.8 (14.3-19.7) | 11.2 (8.9-13.7) | 12.6 (10.4-15.1) |
| | South | 1,425 | 65.7 (61.9-69.4) | 12.0 (9.9-14.6) | 9.5 (7.5-11.9) | 12.7 (10.4-15.5) |
| Living alone | Yes | 1,516 | 48.7 (44.6-52.9) | 20.2 (17.3-23.6) | 14.3 (11.8-17.2) | 16.8 (14.1-19.8) |
| | No | 3,385 | 67.3 (65.0-69.6) | 13.2 (11.8-14.8) | 8.8 (7.5-10.2) | 10.7 (9.3-12.2) |
| Social support | Poor | 473 | 52.9 (45.8-59.9) | 14.9 (11.1-19.9) | 16.8 (12.1-23.0) | 15.3 (11.6-20.0) |
| | Moderate | 2,153 | 60.7 (57.4-63.8) | 15.7 (13.5-18.1) | 10.7 (8.9-12.8) | 13.0 (11.1-15.2) |
| | Strong | 2,153 | 65.5 (62.5-68.5) | 15.4 (13.3-17.6) | 8.2 (6.8-9.9) | 10.9 (9.1-13.0) |

[a]Weighted prevalences and 95% confidence intervals (95% CIs)

### Prevalence of SRH, SRMH, depressive symptoms and anxiety symptoms in relation to T2D risk and age group

A total of 72.0% (95% CI 69.9–74.0) of adults rated their health as very good or good and 40.4% (95% CI 38.2–42.7) rated their mental health as excellent or very good. 14.4% (95% CI 12.7–16.3) had depressive symptoms and 11.0% (95% CI 9.5–12.7) had anxiety symptoms.

Table 2 shows the prevalence of SRH, SRMH, depressive symptoms and anxiety symptoms across categories of T2D risk. The prevalence of very good or good SRH as well as of excellent or very good SRMH decreased from the lowest risk category (82.4% or 45.6%, respectively) to the highest T2D risk category (46.6% and 28.4%, respectively). The prevalence of depressive symptoms increased from the category of very low T2D risk (12.5%) to the category of elevated T2D risk (21.1%) and stagnated in the category of high T2D risk (17.2%). The prevalence of anxiety symptoms did not significantly differ between T2D risk categories.

The additional analysis on the prevalence of the health outcomes across age groups indicates that general and mental health were rated worse with increasing age group (S2 Table). In contrast, the prevalence of depressive symptoms and anxiety symptoms decreased with increasing age group.

### Association between T2D risk and SRH, SRMH, depressive symptoms and anxiety symptoms

Table 3 shows the results of Poisson regression analyses separately for each health outcome. Without any adjustment (model 1), participants with a low T2D risk, with an elevated T2D risk and with a high T2D risk were each less likely to report very good or good general health compared to those at very low T2D risk. After adjustment for sex, there was only a minimal change in the prevalence ratios (model 2). The association was slightly attenuated, but persisted after further

**Table 2. Prevalence (95% CI) of self-rated health, self-rated mental health, depressive symptoms and anxiety symptoms by categories of T2D risk among adults without diabetes (n = 4,909).**

| | Categories of T2D risk | | | | |
| | Very low risk (<2%) | Low risk (2% to <5%) | Elevated risk (5% to <10%) | High risk (≥10%) | |
| | % (95% CI)[a] | % (95% CI)[a] | % (95% CI)[a] | % (95% CI)[a] | p-value |
|---|---|---|---|---|---|
| Very good/good self-rated health | 82.4 (79.9-84.8) | 60.6 (55.3-65.5) | 59.8 (53.3-65.9) | 46.6 (41.1-52.3) | <0.001 |
| Excellent/very good self-rated mental health | 45.6 (42.6-48.8) | 36.3 (31.7-41.1) | 31.0 (26.0-36.6) | 28.4 (23.5-33.9) | <0.001 |
| Depressive symptoms | 12.5 (10.5-14.9) | 15.3 (11.3-20.4) | 21.1 (15.1-28.8) | 17.2 (13.2-22.1) | 0.017 |
| Anxiety symptoms | 10.2 (8.3-12.4) | 9.4 (6.2-13.8) | 14.4 (9.3-21.6) | 13.9 (10.1-18.7) | 0.192 |

Missing values: self-rated health (n = 1), self-rated mental health (n = 12), depressive symptoms (n = 80), anxiety symptoms (n = 60) [a]Weighted prevalences and 95% confidence intervals (95% CIs)

**Table 3. Prevalence ratio (95% CI) for the association of T2D risk with self-rated health, self-rated mental health, depressive symptoms and anxiety symptoms among adults without diabetes (n = 4,909).**

| | n | Categories of T2D risk | | | |
| | | Very low risk (<2%) | Low risk (2% to <5%) | Elevated risk (5% to <10%) | High risk (≥10%) |
| | | PR (95% CI)[a] | PR (95% CI)[a] | PR (95% CI)[a] | PR (95% CI)[a] |
|---|---|---|---|---|---|
| **Very good/good self-rated health (SRH)** | | | | | |
| Model 1 | 4,908 | 1 (reference) | 0.73 (0.67-0.80) | 0.72 (0.65-0.81) | 0.57 (0.50-0.64) |
| Model 2 | 4,908 | 1 (reference) | 0.73 (0.67-0.80) | 0.72 (0.65-0.80) | 0.56 (0.49-0.63) |
| Model 3 | 4,908 | 1 (reference) | 0.78 (0.70-0.86) | 0.78 (0.68-0.88) | 0.61 (0.53-0.70) |
| Model 4 | 4,762 | 1 (reference) | 0.82 (0.74-0.90) | 0.83 (0.73-0.94) | 0.65 (0.56-0.75) |
| **Excellent/very good self-rated mental health (SRMH)** | | | | | |
| Model 1 | 4,897 | 1 (reference) | 0.79 (0.69-0.92) | 0.68 (0.56-0.82) | 0.62 (0.51-0.76) |
| Model 2 | 4,897 | 1 (reference) | 0.78 (0.67-0.90) | 0.67 (0.56-0.80) | 0.60 (0.49-0.73) |
| Model 3 | 4,897 | 1 (reference) | 0.76 (0.64-0.90) | 0.65 (0.53-0.80) | 0.58 (0.46-0.73) |
| Model 4 | 4,752 | 1 (reference) | 0.84 (0.71-0.98) | 0.72 (0.59-0.88) | 0.65 (0.51-0.81) |
| **Depressive symptoms** | | | | | |
| Model 1 | 4,829 | 1 (reference) | 1.22 (0.87-1.72) | 1.69 (1.17-2.44) | 1.38 (1.01-1.88) |
| Model 2 | 4,829 | 1 (reference) | 1.24 (0.88-1.74) | 1.70 (1.18-2.46) | 1.41 (1.03-1.93) |
| Model 3 | 4,829 | 1 (reference) | 2.19 (1.49-3.23) | 3.19 (2.12-4.80) | 3.04 (2.09-4.43) |
| Model 4 | 4,700 | 1 (reference) | 2.11 (1.47-3.02) | 2.59 (1.74-3.87) | 2.48 (1.70-3.63) |
| **Anxiety symptoms** | | | | | |
| Model 1 | 4,849 | 1 (reference) | 0.92 (0.59-1.44) | 1.42 (0.89-2.26) | 1.36 (0.95-1.97) |
| Model 2 | 4,849 | 1 (reference) | 0.95 (0.61-1.48) | 1.46 (0.92-2.32) | 1.47 (1.02-2.12) |
| Model 3 | 4,849 | 1 (reference) | 1.74 (1.03-2.94) | 2.77 (1.61-4.79) | 3.23 (2.03-5.15) |
| Model 4 | 4,708 | 1 (reference) | 1.58 (0.95-2.63) | 2.07 (1.15-3.73) | 2.50 (1.54-4.05) |

Missing values: self-rated health (n = 1), self-rated mental health (n = 12), depressive symptoms (n = 80), anxiety symptoms (n = 60), educational level (n = 12), living alone (n = 8) and social support (n = 130)

[a]Weighted prevalence ratios (PR) and 95% confidence intervals (95% CIs)

PR derived from separate Poisson regression models with self-rated health, self-rated mental health, depressive symptoms and anxiety symptoms as dependent variables. Model 1: unadjusted; Model 2: adjusted for sex; Model 3: adjusted additionally for age; Model 4: adjusted additionally for educational level, region, living alone and social support.

adjusting for age (model 3) and after adjusting for all considered covariates (model 4). In the fully adjusted model, SRH was less likely rated as very good or good among adults with a low risk (PR 0.82, 95% CI 0.74–0.90), an elevated risk (0.83, 0.73–0.94) and a high risk (0.65, 0.56–0.75) compared to adults with very low T2D risk (Table 3; Fig 1).

In the unadjusted model, participants with a low T2D risk, with an elevated T2D risk and with a high T2D risk were less likely to have an excellent or very good SRMH than those at very low T2D risk. Results remained similar after adjustment for sex and for age. Further adjustment slightly attenuated the inverse association, but adults with a low T2D risk (0.84, 0.71–0.98), an elevated risk (0.72, 0.59–0.88) and a high risk (0.65, 0.51–0.81) were still less likely to rate their mental health as excellent or very good than those with a very low T2D risk (Table 3; Fig 1).

Without any adjustment, participants with an elevated or with a high T2D risk were at higher risk for depressive symptoms compared to those at very low T2D risk. Results persisted after adjusting for sex. Additional adjustment for age strengthened the observed association between T2D risk and depressive symptoms and adjustment for the further covariates slightly attenuated the association. In the fully adjusted model, adults with a low T2D risk (2.11, 1.47–3.02), an elevated risk (2.59, 1.74–3.87) and a high T2D risk (2.48, 1.70–3.63) were more likely to report depressive symptoms than those with a very low T2D risk (Table 3; Fig 1).

In the unadjusted model, no association between T2D risk and anxiety symptoms was observed. After adjusting for sex, a significant association between anxiety symptoms and T2D risk was limited to persons with high T2D risk. After further adjustment for age, the association was strengthened and adults with a low, elevated and high T2D risk were more likely to report anxiety symptoms compared to adults with a very low risk of T2D. Results in the fully adjusted model were slightly attenuated, but persisted for adults with an elevated T2D risk (2.07, 1.15–3.73) and a high T2D risk (2.50, 1.54–4.05) (Table 3; Fig 1).

In a sensitivity analysis with imputed covariate and outcome data, results were very similar to those of the main complete-case analysis (S1 Table). In another sensitivity analysis testing for a linear relationship between T2D risk

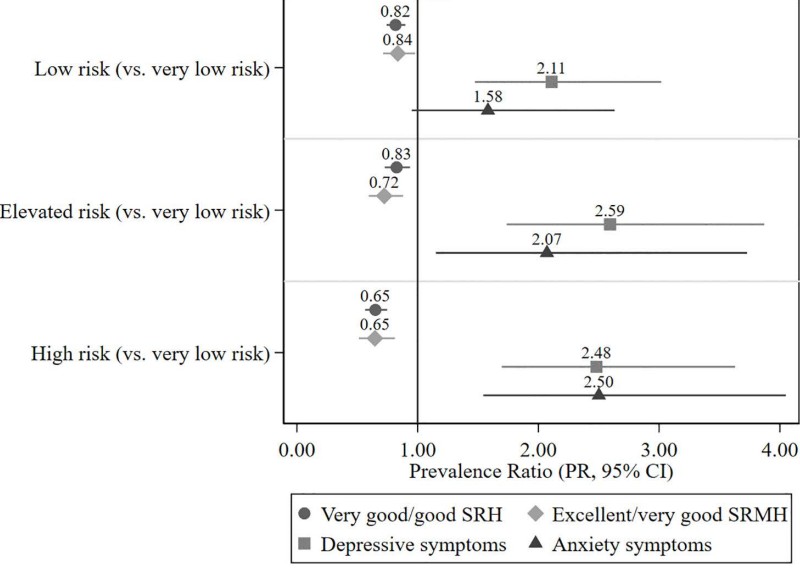

**Fig 1. Forest plot of prevalence ratios (95% CI) for the association of T2D risk with health outcomes.** Shown are weighted prevalence ratios (PRs) with 95% confidence intervals for the association of T2D risk score categories with self-rated health, self-rated mental health, depressive symptoms, and anxiety symptoms among adults without diabetes (n = 4,909). Reference category: very low T2D risk (<2%). Estimates are based on Poisson regression with robust variance estimation, fully adjusted (model 4) for sex, age, educational level, living alone, and social support. Abbreviations: T2D, type 2 diabetes; PR, prevalence ratio; CI, confidence interval; SRH, self-rated health; SRMH, self-rated mental health.

(logarithmically transformed) and health outcomes, results from the main analyses were confirmed. Significant associations between the logarithmically transformed T2D risk and health outcomes were observed in all models except for the unadjusted model with anxiety symptoms (S3 Table).

Further, there was no significant interaction of sex or age with T2D risk (logarithmically transformed) with respect to the health outcomes, except for an interaction of age with T2D risk regarding SRH (S4 Table). Age-stratified analysis revealed that the proportion of adults with very good/good SRH decreased with higher T2D risk among middle-aged (45–64 years) and older (65 years and older) adults, but not among adults under 45 years of age (S2 Table). Finally, the calculation of pairwise Pearson correlations illustrated the suppression effect of age in the association between T2D risk and depressive symptoms as well as between T2D risk and anxiety in (S5 Table), which led to the above mentioned strengthening of the associations when additionally adjusting for age in model 3 (Table 3).

## Discussion

In this population-based study, 60.8% of adults without known diabetes in Germany had a very low predicted 5-year-T2D risk and 15.7% a low risk, whereas 10.7% had an elevated risk and 12.8% a high risk for T2D, similar to a previous study in Germany [39]. In line with our hypothesis, adults with high T2D risk had a 35% lower probability of rating their general health as very good or good or rating their mental health as excellent or very good compared to those with very low T2D risk. Furthermore, participants with a high T2D risk had a 2.5-fold higher probability of having depressive or anxiety symptoms compared to those with a very low T2D risk. For all health indicators, the probability for an unfavourable outcome was also significantly increased among adults with an elevated T2D risk and even among adults with a low T2D risk, except for anxiety symptoms.

While a link between SRH and manifest T2D is well established [5,62,63], epidemiological evidence directly examining validated T2D risk scores, comprising multiple lifestyle-related and biological risk factors, in relation to SRH in current nationwide population samples remains limited. In support of the observed association between higher T2D risk and poorer SRH, few previous observational studies showed associations between behavioural risk factors and worse SRH [26,27] or between protective factors and good SRH [28].

Additionally, an association between higher T2D risk and lower levels of SRMH was observed in our study. To the best of our knowledge, no previous studies have examined the association between this particular mental health indicator and T2D risk or healthy lifestyle scores. Previous health surveys in the US, Germany, and Belgium consistently found strong associations between unhealthier lifestyles and poorer mental health, operationalized by frequent mental distress, vitality, or life satisfaction [25,29,64]. Based on cross-sectional population-based data from Germany, a robust association of perceived chronic stress with predicted 5-year-T2D risk was found, also using the GDRS [65].

Our findings of a positive association between T2D risk and depressive symptoms are in line with previous studies reporting that those with healthy behaviours have a lower risk of depression or depressive symptoms [24,29]. A nationwide cohort study from Germany showed that particularly women with depression and prediabetes had a considerably lower chance for remission to normoglycaemia over time and were more likely to maintain their pre-existing elevated T2D risk compared to those without depression [66]. There is increasing evidence for a bidirectional relationship between T2D risk or cardiovascular risk and depression, though the underlying mechanisms remain unclear [9,67,68].

We also found a positive association between T2D risk and anxiety symptoms, albeit only after adjusting for age. This can be explained by a suppression effect [61] owing to the fact that T2D risk and age are associated, but correlate to prevalent anxiety symptoms in opposite directions. Previous studies support an association between T2D risk and anxiety. Recent reviews and meta-analyses found that anxiety symptoms or disorders increase the risk of T2D [10,11], with one study showing this effect particularly in adults with prediabetes [69]. While it has been less clear so far whether combined lifestyle or T2D risk scores predict anxiety symptoms or disorders, recent evidence from a meta-analysis of lifestyle interventions supports this association [70].

In the present study, we observed no significant interactions between sex and T2D risk, indicating that the association between T2D risk and the health outcomes did not differ significantly between men and women. However, a significant interaction between T2D risk and age was found with regard to SRH, suggesting that a higher T2D risk was not associated with worse SRH in young adults. It should be noted that the number of cases of younger participants with a high T2D risk was very low in this study, which is why this observed interaction must be interpreted with caution. The present analysis showed that the risk of depressive and anxiety symptoms in participants with a high T2D risk was slightly reduced after adjustment for living alone and social support. Previous studies also stated that a high level of social support is associated with reduced T2D symptoms as well as reduced depressive symptoms [71] and a low level of social support is associated with more depressive symptoms [72].

While results from previous observational studies clearly point to a bidirectional relationship between lifestyle-related or cardiometabolic risk profiles and various indicators of self-rated general health, mental health or common mental disorders, the underlying mechanisms and the causal pathways remain to be elucidated [67,73]. Besides a behavioural pathway, shared biological mechanisms associated with the endocrinological, metabolic and immunological systems could be one possible reason for the bidirectional link [6,74–76].

## Public health relevance and implications for future research

T2D, general and mental health problems and their interrelationship lead to a high burden of disease in the population and are therefore of great relevance to public health [77]. Results of the present analysis showed that not only those with a high T2D risk, but also those with a low and elevated T2D risk have a lower probability of a good SRH or SRMH and a higher risk of depressive and anxiety symptoms. In light of previous research supporting a strong and most likely bidirectional link between T2D risk and mental health, our findings underline that physical and mental health should be considered together in the context of prevention and health promotion strategies.

The complex link and causal pathways between mental health and T2D risk is not fully understood, highlighting the need for more longitudinal studies. However, evidence from intervention studies shows that addressing T2D risk factors and improving physical health may also have a beneficial impact on other health outcomes, including health-related quality of life [78–80], and depressive and anxiety symptoms [70,81]. Importantly, there is also evidence that lifestyle interventions need to integrate mental health in order to be effective. Behavioural changes are essential for reducing T2D risk [82], but basic lifestyle advice alone is not effective [83]. Future public health research needs to put a focus on harnessing primary prevention strategies synergistically beneficial to the promotion of physical and mental health of the population [69,84].

Clinical guidelines endorse routine assessment and management of psychosocial problems in people with diabetes [85–87]. These guidance documents principally address patients with manifest diabetes and clinical care pathways; population-level guidance for persons at elevated T2D risk without disease is sparse. Our findings therefore support considering prevention strategies that concurrently target behavioural risk factors and mental-health outcomes in high-risk, non-diabetic populations.

Longitudinal observational studies are needed to improve our understanding of the underlying mechanisms and to identify high risk groups and the role of contextual factors. Based on the results, community-based or setting-based intervention studies can be designed and evaluated. Essential to both types of studies is the use of validated and established tools to define relevant health indicators as a basis for conducting systematic reviews and meta-analyses [88].

## Strength and limitations

Based on nationwide population-based survey data from GEDA 2022, valid up-to-date information on the association between predicted 5-year-T2D risk and general and mental health is provided. Using the established GDRS instrument as an accessible tool for various settings, several lifestyle-related T2D risk factors have been considered.

There are several limitations. Most importantly, the cross-sectional analysis does not allow any conclusions to be drawn regarding the direction of the relationship between T2D risk and mental health, let alone the underlying causal pathways. However, in conjunction with previous research from observational and intervention studies pointing to a bidirectional link between T2D risk and mental health, our results highlight the need to address the association in primary prevention and health promotion. Furthermore, the GEDA study is a telephone survey and has therefore some methodological limitations. Participants with a low educational level, participants with severe (mental) health problems and those living in institutions are likely to be under-represented in this type of study. Although the complex weighting procedure [89] mitigates this problem, the selective participation of respondents should be considered in the interpretation of the results, leading to a potential underestimation of depressive and anxiety symptoms as well as T2D risk. In addition, the interview was conducted in German; therefore, people without sufficient German language skills could not take part in this survey. Due to the fear of stigmatisation of mental health problems and the effect of social desirability, the responses of the participants can be biased. Finally, the GDRS as well as general and mental health indicators only consider self-reported data and no measurement data or clinical diagnoses. Nevertheless, both the GDRS as well as all indicators of general and mental health applied in the present study are established validated tools specifically apt for non-invasive low-threshold population-based studies. Because waist circumference could not be measured, we estimated it from self-reported height, weight, age, and sex using an established equation. However, it is possible that the results would have varied if a measured waist circumference had been available for calculation of the GDRS.

## Conclusion

In this nationwide health survey of adults in Germany, a higher predicted T2D risk was associated with a lower probability of a positive SRH and SRMH as well as a higher probability of depressive and anxiety symptoms. These associations were independent of sociodemographic and social determinants.

In conjunction with a large body of research indicating a bidirectional association between T2D risk and mental health with potentially underlying causal pathways, these findings highlight the need and the potential to address two major current public health challenges synergistically. Further research from prospective and intervention studies is urgently needed to improve insight into the complex interaction of T2D risk and mental health and to provide proof of principle of how both components can be strengthened over the life course.

### Declaration of generative AI in scientific writing

During the preparation of the final manuscript the author used ChatGPT and DeepLWrite in order to shorten and improve the readability and language of individual sentences. After using this tool, the author reviewed and edited the content as needed and takes full responsibility for the content of the published article.

### Supporting information

**S1 Table. Prevalence ratio (95% CI) for the association of T2D risk with self-rated health, self-rated mental health, depressive symptoms, and anxiety symptoms among adults without diabetes, multiple imputation (n=4,909).** Missing values: self-rated health (n=1), self-rated mental health (n=12), depressive symptoms (n=80), anxiety symptoms (n=60), educational level (n=12), living alone (n=8) and social support (n=130). [a] Weighted prevalence ratios (PR) and 95% confidence intervals (95% CIs) were derived from separate Poisson regression models with self-rated health, self-rated mental health, depressive symptoms and anxiety symptoms as dependent variables. Model 4: adjusted additionally for educational level, region, living alone and social support.
(DOCX)

**S2 Table. Prevalence (95% CI) of self-rated health, self-rated mental health, depressive symptoms and anxiety symptoms by age among adults without diabetes (n = 4,909).** Missing values: self-rated health (n = 1), self-rated mental health (n = 12), depressive symptoms (n = 80), anxiety symptoms (n = 60).
(DOCX)

**S3 Table. Prevalence ratio (95% CI) for the association of continuous T2D risk with self-rated health, self-rated mental health, depressive symptoms, and anxiety symptoms among adults without diabetes (n = 4,909).** Missing values: self-rated health (n = 1), self-rated mental health (n = 12), depressive symptoms (n = 80), anxiety symptoms (n = 60), educational level (n = 12), living alone (n = 8) and social support (n = 130). [1] Weighted prevalence ratios (PR) and 95% confidence intervals (95% CIs) were derived from separate Poisson regression models with self-rated health, self-rated mental health, depressive symptoms and anxiety symptoms as dependent variables. Model 1: unadjusted; Model 2: adjusted for sex; Model 3: adjusted additionally for age; Model 4: adjusted additionally for educational level, region, living alone and social support.
(DOCX)

**S4 Table. Interaction of continuous T2D risk score with sex and age for self-rated health, self-rated mental health, depressive symptoms, anxiety symptoms and among people without diabetes (Prevalence ratio and 95% CI) (n = 4,909).** Missing values: self-rated health (n = 1), self-rated mental health (n = 12), depressive symptoms (n = 80), anxiety symptoms (n = 60), educational level (n = 12), living alone (n = 8) and social support (n = 130). [1] p-values were derived from Poisson regression model with self-rated health, self-rated mental health, depressive symptoms and anxiety symptoms as dependent variables. Adjusted for educational level, region, living alone and social support.
(DOCX)

**S5 Table. Pairwise correlations between age, T2D risk and self-rated health, self-rated mental health, depressive symptoms, anxiety symptoms and among people without diabetes (n = 4,909).** *p < 0.05. Missing values: self-rated health (n = 1), self-rated mental health (n = 12), depressive symptoms (n = 80), anxiety symptoms (n = 60), educational level (n = 12), living alone (n = 8) and social support (n = 130).
(DOCX)

## Acknowledgments

We would like to express our sincerest thanks to all participants of the GEDA study. We would like to extend our gratitude to the interviewers at USUMA GmbH and the colleagues of the GEDA team at the RKI for their dedicated cooperation.

## Author contributions

**Conceptualization:** Laura Neuperdt, Yong Du, Christa Scheidt-Nave, Jens Baumert, Christin Heidemann.

**Formal analysis:** Laura Neuperdt, Yong Du, Jens Baumert.

**Funding acquisition:** Julia Nübel, Christa Scheidt-Nave, Jens Baumert, Christin Heidemann.

**Methodology:** Laura Neuperdt, Yong Du, Christa Scheidt-Nave, Jens Baumert, Christin Heidemann.

**Project administration:** Julia Nübel, Christa Scheidt-Nave, Jens Baumert, Christin Heidemann.

**Supervision:** Christin Heidemann.

**Visualization:** Laura Neuperdt, Yong Du, Julia Nübel, Christa Scheidt-Nave, Jens Baumert, Christin Heidemann.

**Writing – original draft:** Laura Neuperdt.

**Writing – review & editing:** Yong Du, Julia Nübel, Ulfert Hapke, Lena Walther, Gert B. M. Mensink, Almut Richter, Christa Scheidt-Nave, Jens Baumert, Christin Heidemann.

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
