## [Decision Letter · Decision Letter 0]

12 Aug 2025

Dear Dr. Neuperdt,

Thank you for submitting your manuscript to PLOS ONE. After careful consideration, we feel that it has merit but does not fully meet PLOS ONE’s publication criteria as it currently stands. Therefore, we invite you to submit a revised version of the manuscript that addresses the points raised during the review process.

**ACADEMIC EDITOR: Major revision**

We look forward to receiving your revised manuscript.

Kind regards,

Marwan Salih Al-Nimer, MD, PhD

Academic Editor

PLOS ONE

Journal Requirements:

2. In the ethics statement in the Methods, you have specified that verbal consent was obtained. Please provide additional details regarding how this consent was documented and witnessed, and state whether this was approved by the IRB

“This work was supported by a grant from the German Center for Diabetes Research (DZD) funded by the German Federal Ministry of Education and Research (grant number: HMGU2022Z5) and grants from the German Federal Ministry of Health within the framework of the National Diabetes Surveillance project at the Robert Koch Institute (grant numbers: 2522DIA700 and 2523DIA002). The survey GEDA 2022 was funded by the Robert Koch Institute and the Federal Ministry of Health in Germany.”

4. We noted in your submission details that a portion of your manuscript may have been presented or published elsewhere. “Parts of the results have previously been presented in abstract form at two scientific conferences (a German-language abstract for a national joint public health conference in Dresden [1], and an English-language poster at the 17th European Public Health Conference [2]). These prior formats do not constitute formal publication, and the full manuscript is original and has not been published or submitted elsewhere.

1. Neuperdt L, Du Y, Nübel J, Hapke U, Walther L, Mensink GBM, et al. Assoziation des Typ-2-Diabetes-Risikos mit selbsteingeschätzter allgemeiner und psychischer Gesundheit bei Erwachsenen in Deutschland – Ergebnisse der Studie GEDA 2022 Gesundheit – gemeinsam Kooperationstagung der Deutschen Gesellschaft für Medizinische Informatik, Biometrie und Epidemiologie (GMDS), Deutschen Gesellschaft für Sozialmedizin und Prävention (DGSMP), Deutschen Gesellschaft für Epidemiologie (DGEpi), Deutschen Gesellschaft für Medizinische Soziologie (DGMS) und der Deutschen Gesellschaft für Public Health (DGPH); Dresden, Germany German Medical Science GMS Publishing House; 2024. doi: 10.3205/24gmds625

2. Neuperdt L, Du Y, Nübel J, Hapke U, Walther L, Mensink GBM, et al. Type 2 diabetes risk is associated with poorer general health and mental health in women and men.  17th European Public Health Conference 2024; Lisbon, Portugal: European Journal of Public Health; 2024. doiI: 10.1093/eurpub/ckae144.1291”

4. You have indicated that data is available from [fdz@rki.de]. Please can we ask you to provide us with a general contact email address for the data requests, so readers can request access in perpetuity. If a general email is not available please provide a link to a website where readers can obtain access to data.

**Additional Editor Comments:**

Dear authors

The following points need explanation

1: Verbal consent is usually unacceptable, and simply will weaken the study. Add a brief sentence that using verbal consent will not weak the results

2: The data of waist circumference was generated from an equation using height, weight, sex, and age (ref. 37). Please add the marginal errors in using this equation

Reviewers' comments:

Reviewer's Responses to Questions

**Comments to the Author**

1. Is the manuscript technically sound, and do the data support the conclusions?

Reviewer #1: Yes

Reviewer #2: Partly

2. Has the statistical analysis been performed appropriately and rigorously?

Reviewer #1: Yes

Reviewer #2: No

3. Have the authors made all data underlying the findings in their manuscript fully available?

Reviewer #1: No

Reviewer #2: Yes

4. Is the manuscript presented in an intelligible fashion and written in standard English?

Reviewer #1: Yes

Reviewer #2: No

Reviewer #1: Higher predicted type 2 diabetes risk is associated with worse mental health and self-rated general health among adults without known diabetes in Germany – Results of the nationwide population-based study GEDA 2022

Telephone interviews of around 5000 german adults. 5-year diabetes risk was determined with German Diabetes Risk Score and self rated health and mental health was examined depending on risk group. Depressive and anxiteysymptoms were more common among those with high risk of diabetes. The study is soundly conducted, but may benefit from further highlighting how it adds to current knowledge.

Comments

Major:

1. Please add a characteristics table (as table 1) stratified by diabetes risk classification.

2. ”We hypothesized that individuals with a higher predicted 5-year T2D risk are less likely to have favourable SRH and SRMH”. This is not tested in Table 3. Instead you test excellent SRH and SRMH. Please harmonize hypothesis and test. Perhaps the outcomes tested in table 3 should be poos SRH and SRMH? Alternatively, that the hypothesis is revised to state that excellent SRH and SRMH are less common among those with high diabetes risk?

3. Please

Minor:

1. It is stated in the introduction that diabetes is characterized by high glucose and caused by insulin resistance. An alternative statement would be that it is defined by high glucose and caused by a combination of life-style and genetic risk-factors. As you later discuss life-style it would be good to also highlight the importance of this factor in the introduction in a similar fashion as you now do for life-style and mental health.

2. You state that you adress a research gap with this paper. Please present what is currently known about this overlap, and also present what current guidelines from EASD and ADA say about addressing mental health in diabetes.

3. This statement in the discussion should be revised: ”epidemiological research on the association of T2D risk, comprising multiple lifestyle-related and biological risk factors, with SRH is scarce.”. There is extensive research in this area, please rephrase or be more specific with objective statements of quantity.

4. Please avoid the use of cardiometabolic health. This is not what you have been studying specifically. There are uncertainties in telephone interviews, predictive instruments and self reported health. Adding another layer of uncertainty by using a loosely defined term such as cardiometabolic health only increases the uncertainty of the study with no added benefit.

Reviewer #2: methods & statistics

1) Clarify the regression specification (robust variance + survey design).

You analyze binary outcomes with Poisson regression to obtain prevalence ratios (PRs). That’s fine if you’re using modified Poisson with a robust (sandwich) variance, ideally in a survey framework. Please state explicitly that you used robust standard errors and how the survey design was set (svyset in Stata: weights, PSUs, strata; variance estimator). At present it only says “the survey procedure” and “Poisson regression” without variance details, which matters for PR CIs and p-values [L159-L167][L171-L179]. See PLOS ONE’s statistical reporting guidance and standard references on modified Poisson.

2) Justify the choice of covariates (risk-score components vs confounders).

The exposure (GDRS) already bundles age, smoking, diet, etc. When you additionally adjust for age (and sometimes variables causally related to lifestyle, e.g., social support), you risk overadjustment or conditioning on mediators. Please include a brief causal rationale (e.g., a DAG) for what you adjust for and why. PLOS encourages transparent confounder rationale in line with STROBE.

3) Dealing with missing data.

Model 4 drops cases with missing education/living alone/social support (n=150; ~3%); please report the model-specific analytic N directly in Table 3 and consider multiple imputation to limit bias if missingness isn’t completely at random [L64-L66][L243-L248].

4) Bivariable vs adjusted inconsistencies (suppression by age).

You explain the anxiety finding as a suppression effect when adjusting for age—good. Consider adding a short supplemental table with correlations between age, GDRS, and outcomes to make this concrete (readers unfamiliar with suppression will appreciate that) [L288-L294]. A one-sentence textbook-style note would help.

5) Distributional reporting of the exposure.

You report the geometric mean of the predicted 5-year risk (1.15%). For readers, the median (IQR) and a simple histogram of predicted risk would be more interpretable; consider adding both. Also, clearly state the scale of the log-transform used in sensitivity analyses (natural log) and report the effect per 1 SD of log-risk for comparability [L185-L192][L249-L257].

6) Imputed waist circumference inside GDRS.

Waist was estimated from height/weight/age/sex using a prior equation; that’s acceptable, but it can shift risk categorization. Please (a) state the predictive performance (R²/RMSE) of the equation you used, (b) do a sensitivity analysis replacing GDRS with a version that omits waist (or uses BMI) to show robustness, and (c) discuss potential misclassification bias [L121-L127. The GDRS validation and simplified versions are good anchors to cite.

7) Statistical reporting format.

Adopt the journal’s preferred style: report PR (95% CI) with consistent decimals and avoid “95%-CI” and “PR1” notations. PLOS’s stats guidance asks for consistent reporting of regression results and measures of variance. See also PLOS ONE’s submission guidelines (statistical reporting section).

Presentation (tables/“graphs”)

Figures: I don’t see any figures/plots in the main text; the manuscript currently presents results via Tables 1–3. Consider adding a forest plot of adjusted PRs (Model 4) for each outcome with risk categories on the y-axis; this would visually summarize your key findings and is a common, reader-friendly presentation in PLOS ONE.

Tables (specific fixes):

• Table 1—category label inconsistency. The text defines SRH categories as “very good/good” vs “fair/bad/very bad,” but Table 1 uses “Average/poor/very poor.” Please align wording (use “Fair/Bad/Very bad” or “Fair/Poor/Very poor” consistently) [L31-L37][L131-L137][L135-L141][L135-L141].

• Table 1—typo in confidence interval. Region “South” shows 29.7 (27-7–31.8); should be 29.7 (27.7–31.8)[L155-L160].

• Table 2—formatting. Use 95% CI not “95%-CI”; ensure a uniform number of decimals across percentages and CIs, and consider reporting exact p to 3 decimals except where p<0.001 [L204-L207].

• Table 3—footnote style. The column header currently shows “PR1 (95%-CI)” while footnote 1 explains the models. Change “PR1” in the header to “PR” and keep the superscript/footnote on the table title or at the end (to avoid reading it as a variable name) [L241-L248][L245-L248].

• Model Ns. Provide the analytic N for each model row block (Model 1–4) since excluding covariate missings changes sample size [L243-L248].

Line-by-line language & style edits (with exact lines)

Note: PLOS doesn’t enforce US vs UK spelling, but they want consistent English throughout. You currently mix “behavioural/favourable/organisation/generalised/anonymised” with American variants. I suggest standardizing to one dialect across the whole paper (examples below). PLOS also prefers “95% CI,” sentence case titles, and clean abbreviation definitions.

Title & front matter

• Short title capitalization looks fine. Ensure sentence case for full title (PLOS guideline) [L36-L37].

Keywords (capitalization consistency):

• “anxiety symptoms; Depressive Symptoms; German Diabetes Risk Score; Population-based survey; Subjective health” → make consistent lowercase: “Anxiety symptoms; depressive symptoms; German Diabetes Risk Score; population-based survey; subjective health.” [L41-L43].

Abstract

• “telephone-interview survey” → telephone interview survey [L35-L36].

• Risk categories: replace “>=10%” with ≥10% (Unicode symbol) for consistency with journal style; similarly for other inequality signs [L36-L39].

• “Poisson regression was used … adjusting for …” → “… and models were adjusted for …” (active, parallel wording) [L39-L41].

• “95% confidence interval, CI” → “95% confidence interval (CI)” [L42-L44].

Introduction

• “etiologically heterogenous chronic disorder” → heterogeneous [L46-L51].

• Consistency: Decide on behavioral vs behavioural and use one form; e.g., “behavioral-related” rather than “behavioural-related” (several instances) [L77-L84].

• Minor: “e. g.” → e.g. (no spaces) [L65-L72].

Methods—Design & participants

• “German speaking resident adult population” → German-speaking [L87-L93].

• “computer-assisted telephone interview (CATI)” already defined—good.

• Ethics tense/consistency: “All participants … must give their verbal consent …” → “… gave their verbal consent …” (past tense) [L101-L107].

• “anonymised transfer” → anonymized (if choosing US spelling) [L103-L107].

• GDPR citation format: “Art. 7 No. 1” → consider “Article 7(1)” for standard legal style [L107-L110].

Methods—Exposure & outcomes

• SRH operationalization: consistent with WHO/MEHM; keep wording aligned with the categories you actually use in tables (see Table 1 point) [L129-L137].

• SRMH: “combined to” → combined into; and your second category title uses “good/fair/poor” while the original response option is “bad”—use “good/fair/bad” or stick to “good/fair/poor” but then reflect that in the prompt text; be consistent across text and tables [L31-L37].

• PHQ-2 / GAD-2 cut-offs: prefer “cutoff” (noun) or “cut-off”; you currently use “cut off” (two words) in several places [L53-L55][L61-L63].

• “Generalised Anxiety Disorder-2 (GAD-2)” → Generalized if adopting US spelling; the instrument name is commonly styled “Generalized Anxiety Disorder-2–item (GAD-2)” in US journals [L57-L61].

Methods—Statistics

• “Rao-Scott-Chi-Square test” → Rao–Scott chi-square test (en dash; lower-case chi-square) [L166-L170].

• “Prevalence Ratios (PR)” → prevalence ratios (PRs); keep pluralization consistent [L171-L179].

• “We utilise Poisson regression …” → utilize or simply use; avoid mixed dialects [L171-L179].

• Spell out design elements of svy (weights/PSU/strata, variance estimator) as noted above.

Results

• Minor discrepancy: text says “A total of 71% self-rated …” while Table 1 shows 72.0%. Harmonize (rounding) [L189-L192][L173-L180].

• Reporting style: keep a uniform decimal precision across proportions and CIs in text (e.g., one decimal for percentages; two for CIs), and always as % (95% CI: a–b) [L197-L205].

Tables

• Table 1: fix CI typo 29.7 (27.7–31.8) for South region [10:L155-L160].

• Table 2 & 3: Use 95% CI (no hyphen) and avoid “PR1” in headers; place the footnote marker after the table title instead [L204-L207][L241-L248].

Discussion

• Great that you note suppression and bidirectionality. Consider adding one citation on modified Poisson/log-binomial vs logistic for PRs to help readers less familiar with PR estimation in cross-sectional data.

One-page list of quick textual fixes (by line)

• Heterogenous → heterogeneous [11:L46-L51]

• telephone-interview survey → telephone interview survey [16:L35-L36]

• 95%-CI → 95% CI (everywhere)[12:L204-L207]

• PR1 (95%-CI) → PR (95% CI) (Table 3 header)[14:L241-L248]

• >=10% → ≥10% (risk thresholds)[16:L36-L39]

• must give … consent → gave … consent (past tense)[2:L101-L107]

• anonymised → anonymized (or make all spellings UK; be consistent)[2:L103-L107]

• e. g. → e.g.[13:L65-L72]

• Generalised → Generalized (if US spelling)[1:L57-L61]

• cut off → cutoff / cut-off (noun)[1:L53-L55][1:L61-L63]

• Rao-Scott-Chi-Square → Rao–Scott chi-square[9:L166-L170]

• behavioural-related → behavioral-related (or standardize to UK throughout)[13:L77-L84]

• SRMH categories: ensure text and tables match (avoid “Average/poor/very poor” if the described options are “fair/bad/very bad”)[1:L31-L37][10:L131-L137]

• Table 1 South CI typo: 29.7 (27.7–31.8)[10:L155-L160]

• 71% vs 72% SRH prevalence: harmonize rounding[9:L189-L192][10:L173-L180]

Bottom-line assessment

• Strengths: Timely national surveillance topic; appropriate use of a validated risk score (GDRS) in a large, weighted sample; consistent inverse associations for SRH/SRMH and positive associations for depressive/anxiety symptoms; thoughtful note on suppression by age.

• Main revisions needed: Fully specify regression variance/design, tighten reporting to PLOS style, fix a few table/text inconsistencies/typos, and add small sensitivity analyses (especially around the imputed waist component of GDRS and exposure scaling). A compact forest plot would also upgrade readability.

**Do you want your identity to be public for this peer review?** For information about this choice, including consent withdrawal, please see our Privacy Policy

Reviewer #1: No

Reviewer #2: No

---

## [Author Response · Author response to Decision Letter 1]

16 Oct 2025

Reviewers' comments:

Reviewer's Responses to Questions

Comments to the Author

1. Is the manuscript technically sound, and do the data support the conclusions?

Reviewer #1: Yes

Reviewer #2: Partly

2. Has the statistical analysis been performed appropriately and rigorously?

Reviewer #1: Yes

Reviewer #2: No

3. Have the authors made all data underlying the findings in their manuscript fully available?

Reviewer #1: No

Reviewer #2: Yes

4. Is the manuscript presented in an intelligible fashion and written in standard English?

Reviewer #1: Yes

Reviewer #2: No

5. Review Comments to the Author

Please use the space provided to explain your answers to the questions above. You may also include additional

comments for the author, including concerns about dual publication, research ethics, or publication ethics. (Please upload your review as an attachment if it exceeds 20,000 characters)

Reviewer #1:

Higher predicted type 2 diabetes risk is associated with worse mental health and self-rated general

health among adults without known diabetes in Germany – Results of the nationwide population-based study GEDA 2022

Telephone interviews of around 5000 german adults. 5-year diabetes risk was determined with German Diabetes Risk Score and self rated health and mental health was examined depending on risk group. Depressive and anxiety symptoms were more common among those with high risk of diabetes. The study is soundly conducted, but may benefit from further highlighting how it adds to current knowledge.

Comments

Major:

1. Please add a characteristics table (as table 1) stratified by diabetes risk classification.

Response: Thank you for this helpful suggestion. We added a characteristics table as table 1 stratified by GDRS risk classification (manuscript with tracked changes line 237-238).

2. ”We hypothesized that individuals with a higher predicted 5-year T2D risk are less likely to have favourable SRH and SRMH”. This is not tested in Table 3. Instead you test excellent SRH and SRMH. Please harmonize hypothesis and test. Perhaps the outcomes tested in table 3 should be poos SRH and SRMH? Alternatively, that the hypothesis is revised to state that excellent SRH and SRMH are less common among those with high diabetes risk?

Response: We have revised the hypothesis wording to ensure consistency with the outcomes presented in Table 3. The hypothesis now states: “We hypothesized that individuals with a higher predicted 5-year T2D risk are less likely to have very good/good SRH and excellent/very good SRMH and are more likely to have depressive and anxiety symptoms.”

Minor:

1. It is stated in the introduction that diabetes is characterized by high glucose and caused by insulin resistance. An alternative statement would be that it is defined by high glucose and caused by a combination of life-style and genetic risk-factors. As you later discuss life-style it would be good to also highlight the importance of this factor in the introduction in a similar fashion as you now do for life-style and mental health.

Response: Thank you for your suggestion. We added the following sentence in the introduction: The pathogenesis of T2D involves a complex interplay of genetic predisposition, lifestyle and environmental factors (Zheng et al, 2018; Ley et al., 2016) (manuscript with tracked changes lines 54-55).

References:

Zheng Y, Ley SH, Hu FB. Global aetiology and epidemiology of type 2 diabetes mellitus and its complications. Nat Rev Endocrinol. 2018;14(2):88–98. Epub 20171208. doi: 10.1038/nrendo.2017.151.

Ley SH, Ardisson Korat AV, Sun Q, Tobias DK, Zhang C, Qi L, et al. Contribution of the Nurses' Health Studies to Uncovering Risk Factors for Type 2 Diabetes: Diet, Lifestyle, Biomarkers, and Genetics. Am J Public Health. 2016;106(9):1624–30. Epub 20160726. doi: 10.2105/ajph.2016.303314.

2. You state that you adress a research gap with this paper. Please present what is currently known about this overlap, and also present what current guidelines from EASD and ADA say about addressing mental health in diabetes.

Response: Regarding the research gap, we have clarified the evidence summary (manuscript with tracked changes lines 69-71) and now state explicitly that, while prior work has linked individual lifestyle-related factors to self-rated health and mental health and a few studies considered combinations of lifestyle factors and self-rated health as well as chronic stress, population-based analyses relating an overall risk profile of multiple health-related factors combined to self-rated health, self-rated mental health and to depressive and anxiety symptoms have not been reported.

Irrespective of this, we agree that guidelines are relevant to the broader topic of mental health in diabetes care. Therefore, we have also added a concise summary of current ADA / ADA–EASD recommendations on psychosocial care to the Discussion (see new paragraph: manuscript with tracked changes lines 407-412), and explicitly note that these guidelines primarily target persons with manifest diabetes; we then discuss the implication that evidence-based guidance for psychosocial screening or interventions in high-risk, but non-diabetic populations remains limited. References have been added (ADA psychosocial position statement; ADA Standards of Care; ADA–EASD consensus).

3. This statement in the discussion should be revised: ”epidemiological research on the association of T2D risk,

comprising multiple lifestyle-related and biological risk factors, with SRH is scarce.”. There is extensive research in this area, please rephrase or be more specific with objective statements of quantity.

Response: Thank you for this suggestion. We revised the text to improve precision and now state: While a link between SRH and manifest T2D is well established [5, 62, 63], epidemiological evidence directly examining validated T2D risk scores, comprising multiple lifestyle-related and biological risk factors, in relation to SRH in current nationwide population samples in Germany remains limited.

4. Please avoid the use of cardiometabolic health. This is not what you have been studying specifically. There are uncertainties in telephone interviews, predictive instruments and self reported health. Adding another layer of uncertainty by using a loosely defined term such as cardiometabolic health only increases the uncertainty of the study with no added benefit.

Response: We thank the reviewer for this valuable suggestion. We have avoided the term “cardiometabolic health” throughout the manuscript as far as possible and replaced it with more precise wording (either “physical health” or “T2D risk”, depending on the context).

Reviewer #2:

methods & statistics

1) Clarify the regression specification (robust variance + survey design).

You analyze binary outcomes with Poisson regression to obtain prevalence ratios (PRs).

That’s fine if you’re using modified Poisson with a robust (sandwich) variance, ideally in a survey framework. Please state explicitly that you used robust standard errors and how the survey design was set (svyset in Stata: weights, PSUs, strata; variance estimator). At present it only says “the survey procedure” and “Poisson regression” without variance details, which matters for PR CIs and p-values [L159-L167][L171-L179]. See PLOS ONE’s statistical reporting guidance and standard references on modified Poisson.

Response: Thank you for this helpful comment. We have now stated the variance details explicitly. The survey design was specified in STATA as svyset [pweight = wQS_tmod]. wQS_tmod denoting the GEDA wave- and module-specific weight. Primary sampling units and strata were not part of the design of this telephone survey, so it was not necessary to account for clustering and stratification. Variances were estimated using Taylor linearization (Stata’s survey default). For bivariate associations we used the Rao–Scott chi-square test. For multivariable analyses, we estimated prevalence ratios (PRs) using modified Poisson regression with a log link and design-based (svy) variance estimation, i.e., Poisson models with robust/sandwich standard errors in a survey framework, as recommended in Stata manual for Poisson regression (StataCorp. Poisson regression (Stata-Manuals; Release 13). 2013 [cited 18.09.2025]. Available from: https://www.stata.com/manuals13/rpoisson.pdf.)

2) Justify the choice of covariates (risk-score components vs confounders).

The exposure (GDRS) already bundles age, smoking, diet, etc. When you additionally adjust for age (and sometimes variables causally related to lifestyle, e.g., social support), you risk overadjustment or conditioning on mediators. Please include a brief causal rationale (e.g., a DAG) for what you adjust for and why. PLOS encourages transparent confounder rationale in line with STROBE.

Response: Following this suggestion, we constructed a DAG to prespecify confounder control. To keep the manuscript concise, we provide the DAG in this response letter rather than as supplementary material; however, we are happy to move it to the Supplement if the reviewer and editor prefer. We treated the GDRS as a composite exposure and therefore did generally not adjust for its individual components to avoid over adjustment; instead, the primary adjustment focused on sociodemographic factors. Because age is part of the GDRS, but is associated with depressive symptoms and anxiety in an opposite direction compared to the GDRS, we implemented a stepwise model sequence with adding age not before Model 3 specifically to assess how age adjustment changes the GDRS-outcome association. The DAG was used solely to structure confounder control in associational analyses, and we refrain from causal claims given the cross-sectional design and potential bidirectionality. We added a sentence in the manuscript (manuscript with tracked changes lines 193-197).

(Please find the figure in the original letter “Response to the Reviewer”.)

Fig. Directed acyclic graph (DAG) used to prespecify covariate adjustment.

The DAG depicts hypothesised relations between the exposure (German Diabetes Risk Score, GDRS), outcomes (self-rated health [SRH], self-rated mental health [SRMH], depressive symptoms, anxiety symptoms), and covariates (sex, age, educational level, region, living alone, social support). Arrows indicate hypothesized directional dependencies used to identify confounding paths and define adjustment sets; they do not imply causation. Because T2D risk and mental health outcomes are bidirectionally related, the DAG encodes a predominant direction solely for adjustment in these cross-sectional analyses. Because age is part of the GDRS, we implemented a stepwise model sequence: after the unadjusted and sex-adjusted models, age was added in Model 3 specifically to assess how age adjustment changes the GDRS-outcome association. DAG created with DAGitty (dagitty.net).

3) Dealing with missing data.

Model 4 drops cases with missing education/living alone/social support (n=150; ~3%); please report the model specific analytic N directly in Table 3 and consider multiple imputation to limit bias if missingness isn’t completely at random [L64-L66][L243-L248].

Response: We thank the reviewer for this valuable suggestion. For transparency, we now report the model-specific analytic N directly in Table 3.

We also conducted multiple imputation for missing values in education, living alone, and social support (manuscript with tracked changes lines 200-203). The imputed models produced results highly consistent with the complete-case analyses. Because the differences between the complete-case and imputed models were negligible, we decided to retain the complete-case results as the main analyses, while providing the imputation results in the Supplementary Materials (S1 Table).

4) Bivariable vs adjusted inconsistencies (suppression by age).

You explain the anxiety finding as a suppression effect when adjusting for age—good. Consider adding a short

supplemental table with correlations between age, GDRS, and outcomes to make this concrete (readers unfamiliar with suppression will appreciate that) [L288-L294]. A one-sentence textbook-style note would help.

Response: We have added a brief supplemental correlation table (S5 Table) showing pairwise Pearson correlations between age, GDRS, and all outcomes. As shown in S5 Table, age is positively correlated with the exposure (GDRS; age–GDRS = 0.619*), while its correlation with the outcome anxiety is slightly negative (age–anxiety = -0.059*), whereas GDRS correlates positively with anxiety (GDRS–anxiety = 0.047*). The same applies to depressive symptoms. We summarized the results if Table S5 in the last sentence of Results and included a one-sentence textbook-style note defining suppression: “A suppression pattern arises when the exposure and a covariate are correlated but relate to the outcome in opposite directions, so that adjusting for the covariate increases the absolute exposure–outcome association.” (manuscript with tracked changes lines 213-215)

5) Distributional reporting of the exposure.

You report the geometric mean of the predicted 5-year risk (1.15%). For readers, the median (IQR) and a simple

histogram of predicted risk would be more interpretable; consider adding both. Also, clearly state the scale of the logtransform used in sensitivity analyses (natural log) and report the effect per 1 SD of log-risk for comparability [L185-L192][L249-L257].

Response:

We now also report the median (1.21%) and IQR (0.27, 4.81%) of the predicted 5-year risk in the Results (manuscript with tracked changes line 222) of the predicted 5-year risk. To keep the manuscript concise, we did not add a figure; a simple histogram is provided here for the reviewer and can be included as Supplemental Figure in the Supplement if the reviewer and editor prefers.

(Please find the figure in the original letter “Response to the Reviewer”.)

Regarding the scaling of the sensitivity analyses, we clarify in lines 211-216 (manuscript with tracked changes) that we used the natural logarithm (ln) applied to the predicted 5-year risk on the probability scale (0–1). For comparability, we additionally estimated associations per 1 SD increase in ln-risk by using a z-standardized ln-risk predictor. Results were very similar to those from the per-unit ln-risk specification (S3 Table). To avoid redundancy, we have not added a separate table to the manuscript or supplement.

(Please find the table in the original letter “Response to the Reviewer”.)

6) Imputed waist circumference inside GDRS.

Waist was estimated from height/weight/age/sex using a prior equation; that’s acceptable, but it can shift risk

categorization. Please (a) state the predictive performance (R²/RMSE) of the equation you used, (b) do a sensitivity analysis replacing GDRS with a version that omits waist (or uses BMI) to show robustness, and (c) discuss potential misclassification bias [L121-L127. The GDRS validation and simplified versions are good anchors to cite.

Response: We thank the reviewer for this careful poi

---

## [Editor Report · Decision Letter 1]

20 Oct 2025

Higher predicted type 2 diabetes risk is associated with worse mental health and self-rated general health among adults without known diabetes in Germany – Results of the nationwide population-based study GEDA 2022

PONE-D-25-38400R1

Dear Dr. Laura Neuperdt,

We’re pleased to inform you that your manuscript has been judged scientifically suitable for publication and will be formally accepted for publication once it meets all outstanding technical requirements.

Kind regards,

Marwan Salih Al-Nimer, MD, PhD

Academic Editor

PLOS ONE
---

## [Editor Report · Acceptance letter]

PONE-D-25-38400R1

PLOS ONE

Dear Dr. Neuperdt,

I'm pleased to inform you that your manuscript has been deemed suitable for publication in PLOS ONE. Congratulations! Your manuscript is now being handed over to our production team.

Kind regards,

on behalf of

Professor Marwan Salih Al-Nimer

Academic Editor

PLOS ONE